# Malocclusion Worsens Survival Following Sepsis Due to the Disruption of Innate and Acquired Immunity

**DOI:** 10.3390/ijms26051894

**Published:** 2025-02-22

**Authors:** Yoshihisa Fujinami, Masafumi Saito, Yuko Ono, Masaya Akashi, Shigeaki Inoue, Joji Kotani

**Affiliations:** 1Department of Emergency Medicine, Kakogawa Central City Hospital, Hyogo 675-8611, Japan; 2Department of Disaster and Emergency and Critical Care Medicine, Graduate School of Medicine, Kobe University, Kobe 650-0047, Japan; windmill@fmu.ac.jp (Y.O.); caf55000@gmail.com (S.I.); kotanijo0412@gmail.com (J.K.); 3Department of Immunology and Microbiology, National Defense Medical College, Saitama 359-8513, Japan; masa9804chicco@gmail.com; 4Department of Oral and Maxillofacial Surgery, Graduate School of Medicine, Kobe University, Kobe 650-0047, Japan; akashimasaya0105@gmail.com; 5Department of Emergency and Critical Care Medicine, Wakayama Medical University, Wakayama 641-0012, Japan

**Keywords:** sepsis, frailty, oral frailty, immunological disruption

## Abstract

Sepsis is a severe condition with high mortality, in which immune dysfunction plays a critical role. Poor oral health has been linked to frailty, but its impact on sepsis outcomes remains unclear. Therefore, we used a mouse model of malocclusion and sepsis to investigate how tooth loss affects immune responses during sepsis. Adult male C57BL/6 mice were divided into four groups: Control, Malocclusion (Mal), Sepsis (CS), and Malocclusion with Sepsis (Mal + CS). Malocclusion was induced by tooth extraction, and sepsis was induced using cecal slurry injection. We assessed survival rates, immune cell counts, and biochemical markers. The Mal + CS group exhibited significantly lower survival rates and greater weight loss compared to the CS group. The flow cytometry showed reduced neutrophils, monocytes, and T cells in the Mal + CS group. Elevated ALT and AST levels indicated liver damage. No significant differences in bacterial loads were observed, but immune suppression was exacerbated in the Mal + CS group. Malocclusion worsens sepsis outcomes by impairing both innate and adaptive immune responses. These findings emphasize the importance of oral health in improving sepsis prognosis and immune function during critical illnesses.

## 1. Introduction

Sepsis is a global health concern. A recent comprehensive clinical study on sepsis revealed 48.9 million cases of sepsis and 11 million deaths in 2017, and surprisingly, this amounts to one in five cases of death globally [1]. Although sepsis is now defined as “sepsis-3” [2], a life-threatening organ dysfunction caused by a dysregulated host response to infection [3], the mechanism underlying this heterogeneous syndrome remains unclear. It is well known that both excessive immune activation and immunosuppression are instantaneously induced by sepsis, which is a pathophysiologic feature of sepsis in the acute phase [4,5]. In addition, Cong et al. showed that an increased neutrophil count was a highly sensitive and specific marker of systemic infections, including sepsis [6]. Thus, these data indicate that investigating the trend of both innate and acquired immune cells is important to improve sepsis outcomes.

Aging is one of the etiologies of immunological dysfunction in the elderly. We have reported that sustained T-cell exhaustion with a reduction in naive T cells and alteration of T-cell phenotypes is associated with poor outcomes in elderly humans and mice [7,8,9]. In addition, chronic inflammation resulting from aging causes immunological dysfunction, which is called inflamm-aging. This state prevents inflammation from moving into a steady state of anti-infection [10]. Furthermore, it was reported that inflamm-aging was associated with frailty [11]. Thus, although one of the confounding factors between critical conditions such as sepsis and frailty might be inflamm-aging in clinical medicine, the directive association or mechanisms are still unknown.

Frailty, defined as a clinically recognizable state of increased vulnerability resulting from an aging-associated decline in reserve and function across multiple physiological systems [12], is the most problematic systemic expression of population aging [13]. A recent meta-analysis showed that frailty in intensive care unit (ICU) patients was independently associated with higher hospital mortality and long-term mortality and with lower rates of discharge home [14]. In addition, it has been shown that physical and cognitive functions are not linearly related to aging [15]. Therefore, frailty is not simply a surrogate marker for aging.

Frailty occurs not only in the whole body but also in oral and swallowing functions. Oral frailty is a new term for the decline in oral function leading to frailty [16]. Poor oral health is clinically evaluated by the number of natural teeth, chewing ability, articulatory oral motor skills, tongue pressure, subjective difficulties in eating tough foods, and subjective difficulty in swallowing [17]. Indeed, an association between poor oral health and frailty has been shown [18]. Furthermore, poor oral health is associated with dementia, including Alzheimer’s disease [19,20,21], cardiovascular diseases [22], diabetes mellitus [23], and chronic obstructive pulmonary disease [24], and strongly predicts the onset of adverse health outcomes, including mortality [25]. In our clinical study, we found that poor oral health with molar malocclusion was independently associated with ADL loss at discharge in ICU patients [26]. We speculate that one of the mechanisms for this is immunological dysfunction. However, little is known about how poor oral health affects the outcomes for acutely ill patients. We speculate that frailty, rather than aging, causes immunological dysfunction and that frailty is caused by oral dysfunction via inflamm-aging.

Therefore, the purpose of this study was to investigate whether poor oral health affects immunological dysfunction and outcomes in septic mice. In this study, we used the tooth loss model to mimic poor oral health, since there is no animal model of poor oral health and oral frailty.

## 2. Results

### 2.1. Significantly Decreased Survival Rate and Body Weight in Septic Mice with Malocclusion

The Control and Mal group mice survived. The 14-day survival rate in the Mal + CS group was significantly lower than that in the CS group (Mal + CS: 50.0%; CS: 75.0%; *p* < 0.05; Figure 1a). The body weight loss of the mice for 24 h after CS injection was significantly lower in the Mal + CS group than in the CS group (Mal + CS: 10.0%; CS: 6.1%; *p* < 0.05; Figure 1b).

### 2.2. Liver Damage Markers, ALT, and AST Were Upregulated in the Plasma of Mal + CS Mice

As shown in Table 1, plasma IL-6 was higher in the CS and Mal + CS groups than in the Control group, and there was no significant difference between the Control and Mal groups or between the CS and Mal + CS groups (*p* < 0.05). In addition, to investigate the function of the liver and kidney in Mal + CS mice, we measured biochemical markers such as aspartate aminotransferase (AST), alanine aminotransferase (ALT), blood urea nitrogen (BUN), and creatinine (CRE). All of these values were higher in CS and Mal + CS than in the Control group. Importantly, CRE and AST were significantly higher in Mal + CS than in the CS group (*p* < 0.05).

### 2.3. Malocclusion Decreased the Number of Neutrophils and NK Cells in the Peripheral Blood in Septic Mice

We investigated the trend of the immune cells 48 h after the induction of sepsis. First, we investigated innate immune cells. To examine the innate immune response in Mal + CS mice against sepsis, we first analyzed the total number and phenotype of neutrophils in the PBMC. As shown in Figure 2, the reduction in neutrophils in PBMC was induced by sepsis (*p* < 0.05).

Interestingly, in the CS group, immature neutrophils significantly increased, while in the Mal + CS group, they decreased (Figure 2b, *p* < 0.05). The number of NK cells tended to be lower in the Mal + CS group than in the CS group, although there was no significant difference between the groups (Figure 2d, *p* = 0.07). Next, we determined the total count of peripheral monocytes. Although the total monocyte counts in Mal + CS tended to be lower than in the CS group, there was no significant difference among each group (Figure 2e). The FACS analysis revealed that highly expressed Ly6C monocytes, defined as inflammatory or immature monocytes, were significantly lower in septic mice (CS and Mal + CS groups) than in non-septic mice (Control and Mal groups) (Figure 2f, *p* < 0.01).

### 2.4. Tooth Extraction Decreased the Number of T Cells in the Peripheral Blood in Septic Mice

To investigate the effect of tooth extraction with or without sepsis on acquired immune cells, we finally analyzed peripheral T and B cells by flow cytometry. As shown in Figure 3, the total number of T cells was significantly lower in Mal + CS than in the CS group (*p* < 0.05).

In addition, the numbers of both CD4 ^+^ and CD8 ^+^ T cells were significantly lower in the Mal + CS group than in the CS group (Figure 3c,d, *p* < 0.05). For B cells, there was no significant difference between the two groups (Figure 3b).

### 2.5. No Difference in Bacterial Colony Counts Between CS and Mal + CS Groups

As shown in Figure 4, when compared between the CS and Mal + CS groups, there was no significant difference in the bacterial loads, neither in the blood nor in the peritoneal fluid. As we anticipated, no bacterial colony was detected in either the Control or Mal group.

## 3. Discussion

In this study, we investigated the effects of malocclusion on sepsis using a mouse model. The Mal + CS group had a significantly lower body weight and survival rate than the CS group. In addition, the Mal + CS group showed a severe reduction in circulating neutrophils and T cells. Because this immune cell reduction was not observed in the Mal group, tooth extraction led to crucial opportunities for providing immune cells, including hematopoiesis and chemotaxis, or for inducing extensive apoptosis of immune cells when inducing sepsis. Our data suggest that tooth extraction increased the severity of sepsis in the acute phase through immune dysfunction.

We showed that in the preparation phase, the malocclusion model and sepsis model worked well: tooth extraction itself did not affect survival, and the CS i.p. caused severe inflammation and organ damage, led to bacteremia, and showed reproducible lethality.

Patients with sepsis show a systemic inflammatory response, often termed the cytokine storm, in the early stage of sepsis and often develop immunosuppression or immunoparalysis, which results in lymphopenia in the late stage [4,27]. It has been reported that the cytokine storm continues for at least 48 h in septic mice [8]. A cytokine storm induces organ blood flow failure and disseminated intravascular coagulation, resulting in multiple organ failure (MOF) [28,29]. In particular, hepatic dysfunction is associated with a worse prognosis than renal or respiratory dysfunction [30]. Therefore, in the present study, it was considered that it was not the continuing immunosuppression but the cytokine storm that caused MOF and caused the mice to die.

Recently, periodontal disease has been reported to lead to systematic inflammation and immunological disruption. However, there is no certain understanding of the mechanisms of the immunological impact, although it has been suspected to engage dendritic cells, macrophages, and T cells [31]. Mouse models of periodontitis developed bacteremia and enhanced the activity of the macrophages [32]. However, whether tooth extraction itself affects the immunological system has not been considered. In the present study, it was certain that the Mal group had oral inflammation, although it was uncertain whether the Mal group had periodontitis; the inflammation persisted for at least 1 week, innate immunity was activated, and acquired immunity was not disrupted.

In this study, the tooth extraction model alone did not yield direct evidence of inflammation or immunosuppression. However, we hypothesize that the tooth extraction led to persistent inflammation and subsequent immunosuppression. Indeed, when assessing the changes from the Control to the CS group (control condition) and from the Mal to the Mal + CS group (malocclusion condition), no significant differences in bacterial load were observed. However, the degree of reduction in neutrophil, monocyte, and T-cell counts was more pronounced in the latter. Regarding the neutrophil response, under the malocclusion condition, the reduction in mature neutrophils was greater, while the increase in immature neutrophils was smaller compared to the control condition. Furthermore, malocclusion resulted in a more substantial decrease in CD4^+^, CD8^+^, and naïve T cells. These findings suggest that tooth extraction, as an intervention, impaired the septic response by affecting both innate and acquired immunity.

In the present study, comparing the Control and Malocclusion groups, malocclusion mice had more neutrophils in their PBMC. Malocclusion mice, 1 week after tooth extraction, were considered to be in an inflammatory condition.

Tooth extraction decreased the survival rate of sepsis patients, and tooth extraction alone resulted in neutropenia and no apparent changes in T and B cell counts. Clinically, neutropenia comorbid with sepsis is known to be more severe [33], indicating that the severity of sepsis was higher in the Mal + CS group. Since it is extremely difficult to adjust the background of immunosuppressed patients due to malignancy or autoimmune diseases in clinical studies, it is a significant finding that the present study suggests that tooth extraction itself is a trigger for immunosuppression or immunoparalysis.

We previously reported that poor oral health affects the functional prognosis of critically ill patients and speculated that immunological dysfunction might be involved in one of the mechanisms [26]. The present study proved that this assumption was valid. Future studies will clarify why tooth extraction triggers immunosuppression and immunoparalysis. We hypothesized that one of these mechanisms was immune disruption caused by odontogenic infection or bacteria that were trapped in the liver. Nagasaki et al. reported that *Porphyromonas gingivalis* (*P.g.*) was identified in the liver of a mouse model of periodontal disease with liver dysfunction [34], and *P.g.* is known to play the role of keystone bacteria, causing local inflammation and systemic immune disturbance [35]. In our study, no bacteria were identified in the Mal group by blood culture, and there was no obvious difference in the number of bacteria between the CS and Mal + CS groups, but we could not deny the possibility of low bacteremia or the possibility of bacteria being trapped in the liver. Unfortunately, we could not prove the presence of *P.g.* in the liver. Confirmation of bacteriological changes in the liver after tooth extraction and an immunological evaluation will be performed in the future.

In addition, although we considered that persistent oral inflammation and successive immunesuppression disrupted the immune response of another infection source, it is unclear whether tooth extraction itself might play an important role. Although we checked that mice after tooth extraction had obvious pneumonia, inflammation models of periodontal or other than oral sources might obtain similar results. We may conduct an experiment using two different infection sources (e.g., lung or abdomen). Thus, we predict that tooth extraction caused oral bacterial infection, because aseptic tooth extraction and keeping the mouse sterile were impossible. Especially in humans, it is plausible that malocclusion and oral infectious inflammation generally coexist, because malocclusion due to periodontitis is a common disease. Although there are some mouse models of periodontitis, it seems to be too difficult to establish them easily [36,37,38]. The impact of tooth loss can be evaluated in humans, because we can intervene in oral care, and we should expand the use of human studies for future work.

## 4. Materials and Methods

### 4.1. Antibodies

All antibodies (Abs) were purchased from Biolegend (San Diego, CA, USA). The following mouse Abs, which were specific to the surface markers, were used: PerCP/Cy5.5-conjugated mouse antiCD4, PerCP/Cy5.5-conjugated antiNK1.1, APC/Cy7-conjugated mouse antiCD8, APC-conjugated mouse antiCD19, APC/Cy7-conjugated mouse antiF4/80, FITC-conjugated mouse antiLy6C, and Pacific blue-conjugated mouse antiLy6G. Mouse Fc-blocker was procured from BD Pharmingen (San Jose, CA, USA). All Abs were diluted to 1/100 in 0.1% bovine serum albumin/phosphate-buffered saline (BSA/PBS).

### 4.2. Animal Experiments

Adult (10–14-week-old) male C57/BL6 mice, weighing 19–22 g, were obtained from CHARLES RIVER LABORATORIES (Yokohama, Japan). All mice were housed in quiet, humidified rooms at 24 °C ± 1 °C with a 12 h light/dark cycle (7 am/7 pm). The mice were allowed access to a standard rodent diet (CE7; Clea, Osaka, Japan) and tap water ad libitum. All experimental protocols were approved by the Institute of Animal Care and Use Committees at Kobe University (No. A190507) and carried out according to the relevant guidelines and regulations.

### 4.3. Malocclusion Model

For establishing the mouse malocclusion model, the maxillary molar teeth of the mice were extracted from both sides using a hook tweezer under isoflurane inhalation, following Takeda et al. [39] with a few modifications. Briefly, the upper molars of the mice were extracted with a hook tweezer while their mouths were kept open with a tweezer. The mice had three teeth, with their upper molars on each side. Considering the width of the hook tweezer, manipulation of tooth extraction was needed 1 or 2 times (Figure 5a). After confirming that the food intake and body weight had recovered to the baseline (Appendix A), CS was injected into the mice to induce sepsis (Figure 5b).

### 4.4. Sepsis Model

We employed the CS model, which is a recent sepsis model established by Wynn et al. [40] and modified by Starr et al. [41]. Briefly, male ICR mice (6–8 weeks old) were sacrificed, and whole cecums were harvested. The mice cecums were snicked and transferred to a nylon-mesh bag, and 1–2 mL of sterile water was poured and filtered twice. The mixture was collected and centrifuged at 11,000 rpm for 1 min. The supernatant was discarded, and the residue was suspended in 30% glycerol at a final concentration of 0.5 mg/mL. CS (400–500 μL) was transferred to cryotubes and stored at −80 °C until use. To induce sepsis, we injected 50 µL of 0.5 mg/mL CS peritoneally into the mice. Thirty minutes after CS administration, the mice were given 800 μL of normal saline solution (according to sepsis for humans: 30 mL/kg) and subcutaneously injected with 0.5 mg of meropenem (according to the treatment for humans: 1 g/50 kg) in their backs. Antimicrobial treatment was administered for 3 days after the induction of sepsis.

### 4.5. Study Design

The mice were randomly divided into four groups: the Control, Malocclusion (Mal), CS, and Mal + CS group. Seven days after tooth extraction, the Control and CS groups were anesthetized for 80 s (equivalent to the operation time) using isoflurane inhalation. The Control and Mal groups of mice were administered 50 μL of 15% glycerol as a vehicle control. The mice were sacrificed 48 h after CS injection for analysis of the serological and bacterial studies. For the survival studies, mice were observed for 14 days after CS injection (Figure 5c).

### 4.6. Sample Collection and Bacterial Colony Counting

Approximately 48 h after CS injection, blood and peritoneal fluid were collected from mice under isoflurane anesthesia. First, a midline incision was made in the abdominal epidermis to expose the peritoneum. The peritoneal cavity was then punctured with a 23G needle (Terumo, Tokyo, Japan), and 100 µL of PBS was introduced to suspend the peritoneal contents for collection. Subsequently, 100 µL of blood was drawn from the inferior vena cava using a 23G needle and transferred into a heparin-coated tube. The collected peritoneal fluid and blood samples were plated on sheep blood agar at a 1:10 dilution and incubated at 37 °C with 5% CO_2_ for 24 h. Bacterial colonies were then visually counted.

### 4.7. Serological Test

The collected blood was centrifuged at 3000 rpm for 15 min at 4 °C. Plasma was collected and stored at −80 °C until use. Blood biochemical parameters, including interleukin (IL)-6, were measured at the Nagahama Institute of Bio-Science and Technology (Shiga, Japan). Liver and renal function were used as serological test parameters to assess sepsis severity, as suggested by Seymour [30].

### 4.8. Flow Cytometric Analysis

Murine peripheral blood mononuclear cells (PBMCs) were isolated by density gradient cell separation using Histopaque 1119 (Sigma-Aldrich, St. Louis, MO, USA). The separated PBMCs were treated with a red blood cell lysis buffer containing 139.5 mM NH_4_Cl and 1.7 mM Tris-HCl (pH 7.65) at 37 °C for 10 min and then washed with 0.1% BSA/PBS. Murine PBMCs were incubated with Abs mixture for 20 min at 4 °C after treatment with mouse Fc-blocker to block nonspecific binding sites. The stained cells were analyzed using FACS Verse (BD Biosciences, Franklin Lakes, NJ, USA). The proportion of the designated cell fraction was determined by recording 10,000 events, and the data files were analyzed using FlowJo software version 10.6.2 (Tree Star Inc. Ashland, OR, USA).

### 4.9. Statistical Methods

The data were analyzed using GraphPad Prism version 8 (GraphPad Software, San Diego, CA, USA). Survival curves were created using the Kaplan–Meier method and compared using the log-rank test. Group differences in the cell counts of peripheral blood and the differences in the immune cells were analyzed using the Mann–Whitney *U*-test and Tukey–Kramer method. *p* < 0.05 was defined as statistically significant. Results are presented as mean ± SD values.

## 5. Conclusions

Malocclusion caused by tooth extraction caused persistent inflammation and successive immunesuppression. In an additional septic state, malocclusion worsened the prognosis because of the disruption of innate and acquired immunity. The implication of this study is that the mechanism by which poor oral health worsens the prognosis of sepsis can be clarified immunologically.

## Figures and Tables

**Figure 1 ijms-26-01894-f001:**
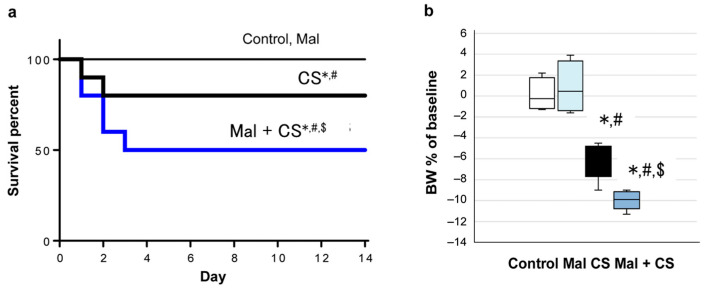
(**a**) Survival curves created with the Kaplan–Meier method. (**b**) The body weight loss of the mice for 24 h after CS injection: *: *p* < 0.05 compared to Control; #: *p* < 0.05 compared to Mal; and $: *p* < 0.05 compared to CS (Control and Mal: *n* = 4; CS and Mal + CS: *n* = 8).

**Figure 2 ijms-26-01894-f002:**
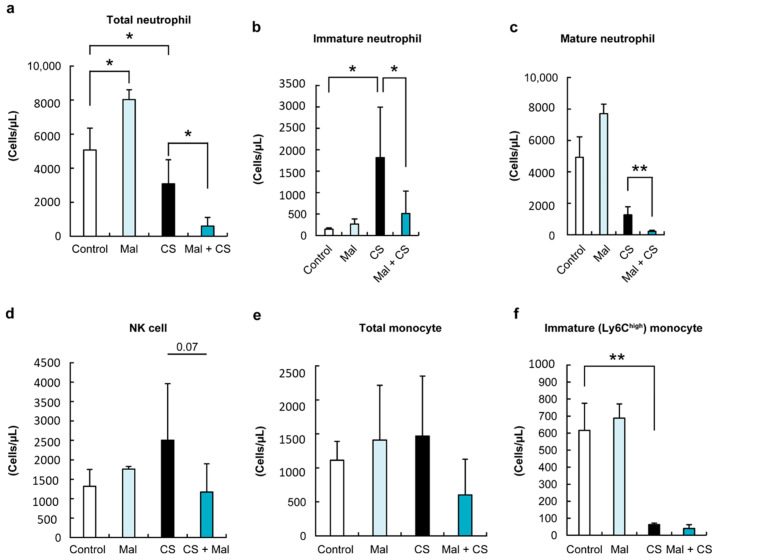
The number of immunological cells (a: total neutrophil; b: immature neutrophil; c: mature neutrophil; d: NK cell; e: total monocyte; and f: immature monocyte). Neutrophils were decreased in the CS and the Mal + CS groups, with lower numbers of immature neutrophils and monocytes in the Mal + CS group compared to the CS group. *: *p* < 0.05, **: *p* < 0.01.

**Figure 3 ijms-26-01894-f003:**
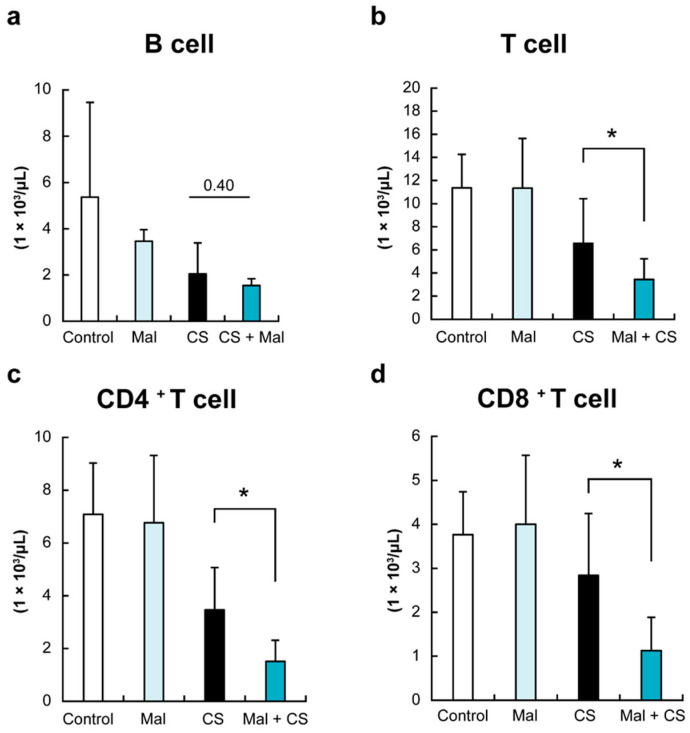
The number of immunological cells (a: B cell; b: T cell; c: CD4 ^+^ T cell; and d: CD8 ^+^ T cell). The numbers of both CD4 ^+^ and CD8 ^+^ T cells were significantly lower in the Mal + CS group than in the CS group *: *p* < 0.05.

**Figure 4 ijms-26-01894-f004:**
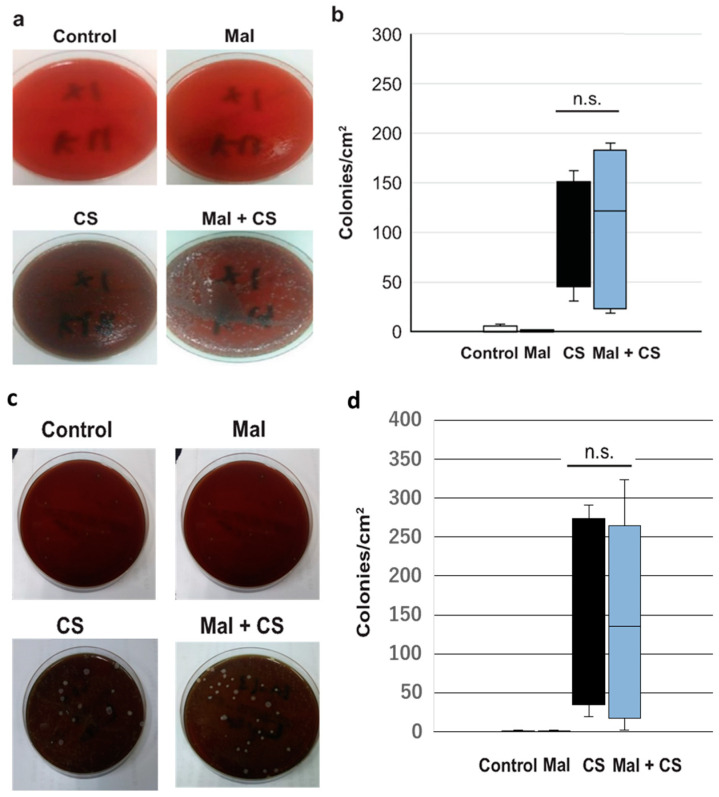
Pictures of the agar medium for blood (**a**) and peritoneal fluid (**c**) culture; graph of the bacterial loads in the blood (**b**) and peritoneal fluid (**d**) culture. n.s.: Not significant.

**Figure 5 ijms-26-01894-f005:**
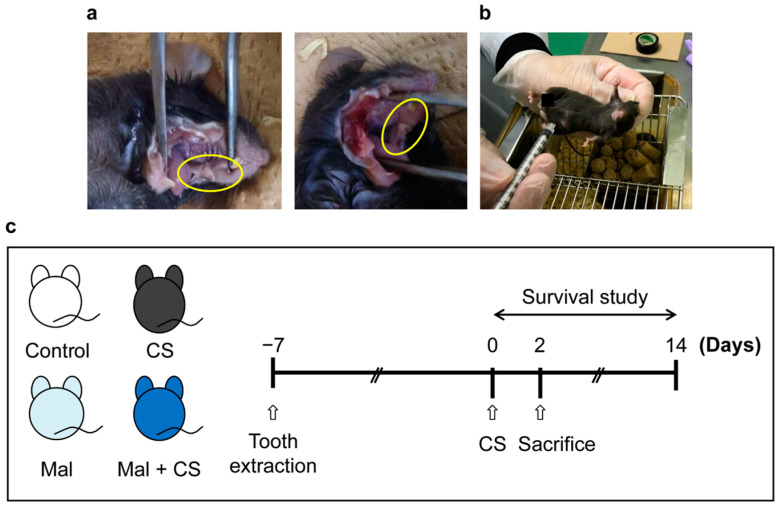
(**a**) Tooth extraction; this picture was taken after sacrifice. The yellow circle on the left indicates the molar region before extraction, while the yellow circle on the right highlights the molar region after extraction. (**b**) Cecal slurry injection. (**c**) Study design.

**Table 1 ijms-26-01894-t001:** Biochemical tests.

	Control	Mal	CS	Mal + CS
IL-6 (pg/mL)	<0.01	<0.01	28.2 ± 23.9 *	32.8 ± 27.4 *
BUN (mg/dL)	30.3 ± 4.0	28.8 ± 5.1	32.1 ± 6.0	34.0 ± 11.6 *
CRE (mg/dL)	0.14 ± 0.03	0.13 ± 0.02	0.16 ± 0.02 *	0.21 ± 0.04 *^,#^
AST (IU/L)	45.8 ± 11.8	45.0 ± 8.3	240.8 ± 55.4 *	322.3 ± 70.6 *^,#^
ALT (IU/L)	28.5 ± 5.7	28.8 ± 4.0	122.0 ± 57.1 *	103.3 ± 46.7 *

IL-6, BUN, creatinine, AST, and ALT were higher in CS and Mal + CS than Control group (* *p* < 0.05). CRE and AST were higher in Mal + CS than CS group (^#^
*p* < 0.05). There was no significant difference between Con and Mal group (*n* = 4–5 per group, Tukey–Kramer method).

## Data Availability

The datasets generated during and/or analyzed during the current study are available from the corresponding author on reasonable request.

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
