# Peer review of "Malocclusion Worsens Survival Following Sepsis Due to the Disruption of Innate and Acquired Immunity"

_ijms, 2025, doi:10.3390/ijms26051894_

Round 1
Reviewer 1 Report
Comments and Suggestions for Authors
Fujinami et al showed the impact of malocclusion on sepsis induced by cecal slurry. Authors showed that tooth extraction when combined with sepsis worsen the survival rates when compared to sepsis. They further showed that malocclusion mediate immune suppression by decreasing the neutrophil, monocytes and T cells counts. However, there is no change in bacterial load in the blood between Mal+CS and CS.
Specific comments: 1) Authors showed bacterial load in the circulation but they should also check it in the peritoneal fluid.
2.) Inflammation induced due to malocclusion can not alone further cause immunosuppression until or unless there is persistent source of infection.
3.) Liver biochemical parameters (AST and CRE) were higher in mal+ CS than CS, can you please provide explanation for this? Why authors have checked the organ damage by biochemical parameters specifically in liver or kidney not in other organs in Mal+CS group vs CS group? Sepsis induced by CS is known to induce organ damage. How malocclusion further worsening this ?
4.) In methods, sepsis model typological error CS (400–500 mL) mL should be uL.
Author Response
[Comment 1] Authors showed bacterial load in the circulation but they should also check it in the peritoneal fluid.
[Response 1] We added the data of peritoneal fluid to Figure 4 and modified Results and Methods.
[Comment 2] Inflammation induced due to malocclusion can not alone further cause immunosuppression until or unless there is persistent source of infection.
[Response 2] As described in previous versions (Line 184–187), we visually assessed the changes from the Control to CS group (control condition) and from the Mal to Mal+CS group (malocclusion condition). The decrease in neutrophils, NK cells, and T cells, as well as reduced exploratory behavior, was more pronounced in the latter condition. Based on these findings, we interpreted this change as an indication that tooth extraction indirectly contributed to immune disturbance. However, as you rightly pointed out, the present study does not provide direct evidence of inflammation or immunosuppression in the tooth extraction model alone. We have now clarified this limitation in the revised manuscript.
[Comment 3] Liver biochemical parameters (AST and CRE) were higher in mal+ CS than CS, can you please provide explanation for this? Why authors have checked the organ damage by biochemical parameters specifically in liver or kidney not in other organs in Mal+CS group vs CS group? Sepsis induced by CS is known to induce organ damage. How malocclusion further worsening this?
[Response 3] Thank you for pointing this out. Liver and renal function were selected as indicators of sepsis severity. Although this approach is not commonly used in clinical medicine, there is a classification of sepsis severity based on liver and renal function, as demonstrated by Seymour et al. (Ref 30).
The observed difference between the CS and Mal+CS groups, despite the absence of apparent organ damage in the Mal group, is considered to reflect progressive multiorgan damage as an indicator of sepsis severity. This interpretation is further supported by the difference in mortality between these groups. In response to your comment, we have added the following clarification in the revised manuscript (Line 304-305).
[Comment 4] In methods, sepsis model typological error CS (400–500 mL) mL should be uL.
[Response 4] Thank you for pointing this out. We have corrected it.
Reviewer 2 Report
Comments and Suggestions for Authors
I reviewed the manuscript "Malocclusion Worsens Survival Following Sepsis Due to the Disruption of the Innate and Acquired Immunity".
In this manuscript, the authors used a mouse model to examine the effects of malocclusion (tooth loss) on sepsis outcomes. The study described the strong controversy that poor oral health is a contributing factor to immunological dysfunction, which makes sepsis worse and reduces survival.
The study has selected an underdeveloped subject in immunology and critical care medicine to determine how oral health is related to sepsis outcomes. The use of a malocclusion model to mimic poor oral health is new, original and makes the results more clinically relevant.
The study has a clear methodology being well-designed to compare control, malocclusion, sepsis, and combined malocclusion-sepsis groups.
The results are accurate, demonstrate that malocclusions exacerbate immune suppression in sepsis, mainly affecting neutrophils, monocytes, and T cells.
The study suggests that oral health is a preventative measure against sepsis related mortality. Thus, in generally there should be more concern on the health of the mouth.
The paper therefore recommends more researches to explore the relationship between oral health and critical illnesses.
This manuscript includes an important and well-executed analysis that contributes to our understanding of the relationship between oral health and sepsis. The results are new and important, and they have clear implications for clinical practice.
I therefore recommend that the paper can be published in actual form.
Author Response
We sincerely appreciate your thoughtful review of our manuscript. We are grateful for your positive evaluation and insightful comments regarding the significance of our study.
We are pleased that you recognized the novelty and clinical relevance of our malocclusion model in investigating the impact of poor oral health on sepsis outcomes. Your acknowledgment of our study’s methodological clarity and the implications of our findings for immunology and critical care medicine is highly encouraging.
We also appreciate your support for further research in this field. We believe that understanding the relationship between oral health and sepsis is crucial for developing preventative strategies against sepsis-related mortality.
Thank you again for your valuable time and thoughtful review. We are honored by your recommendation for publication.
Round 2
Reviewer 1 Report
Comments and Suggestions for Authors
Endorsed